# Antileishmanial Activity of 4,8-Dimethoxynaphthalenyl Chalcones on *Leishmania amazonensis*

**DOI:** 10.3390/antibiotics11101402

**Published:** 2022-10-13

**Authors:** Kaio Maciel de Santiago-Silva, Bruna Taciane da Silva Bortoleti, Laudicéa do Nascimento Oliveira, Fernanda Lima de Azevedo Maia, Joyce Cristina Castro, Ivete Conchon Costa, Danielle Bidóia Lazarin, James L. Wardell, Solange M. S. V. Wardell, Magaly Girão Albuquerque, Camilo Henrique da Silva Lima, Wander Rogério Pavanelli, Marcelle de Lima Ferreira Bispo, Raoni Schroeder B. Gonçalves

**Affiliations:** 1Laboratório de Síntese de Moléculas Medicinais (LaSMMed), Departamento de Química, Centro de Ciências Exatas, Universidade Estadual de Londrina, Londrina 86057-970, PR, Brazil; 2Laboratório de Imunoparasitologia das Doenças Negligenciadas e Câncer (LIDNC), Departamento de Ciências Patológicas, Centro de Ciências Biológicas, Universidade Estadual de Londrina, Londrina 86057-970, PR, Brazil; 3Programa de Pós-Graduação em Biociências e Biotecnologia, Instituto Carlos Chagas (ICC), Fiocruz, Curitiba 81350-010, PR, Brazil; 4Instituto de Química, Universidade Federal do Rio de Janeiro, Rio de Janeiro 21941-909, RJ, Brazil; 5Department of Chemistry, University of Aberdeen, Meston Walk, Old Aberdeen, Aberdeen AB24 3UE, Scotland, UK; 6CHEMSOL, 1 Harcourt Road, Aberdeen AB15 5NY, Scotland, UK

**Keywords:** 4,8-dimethoxynaphthalenyl, arginase, leishmaniasis, neglected diseases, trypanosomatids, trypanothione reductase

## Abstract

Leishmaniasis is a neglected tropical disease caused by *Leishmania* species. Available therapeutic options have several limitations. The drive to develop new, more potent, and selective antileishmanial agents is thus a major goal. Herein we report the synthesis and the biological activity evaluation against promastigote and amastigote forms of *Leishmania amazonensis* of nine 4,8-dimethoxynaphthalenyl chalcones. Compound ((E)-1-(4,8-dimethoxynaphthalen-1-yl)-3-(4-nitrophenyl)prop-2-en-1-one), **4f**, was the most promising with an IC_50_ = 3.3 ± 0.34 μM (promastigotes), a low cytotoxicity profile (CC_50_ = 372.9 ± 0.04 μM), and a high selectivity index (SI = 112.6). Furthermore, **4f** induced several morphological and ultrastructural changes in the free promastigote forms, loss of plasma membrane integrity, and increased reactive oxygen species (ROS). An in silico analysis of drug-likeness and ADME parameters suggested high oral bioavailability and intestinal absorption. Compound **4f** reduced the number of infected macrophages and the number of amastigotes per macrophage, with an IC_50_ value of 18.5 ± 1.19 μM. Molecular docking studies with targets, ARG and TR, showed that compound 4f had more hydrogen bond interactions with the ARG enzyme, indicating a more stable protein-ligand binding. These results suggest that 4,8-dimethoxynaphthalenyl chalcones are worthy of further study as potential antileishmanial drugs.

## 1. Introduction

The leishmaniases are a group of neglected diseases widely distributed among tropical and subtropical countries. Leishmaniases are caused by the protozoan parasite of the genus *Leishmania* and transmitted to humans by the bite of infected female phlebotomine sandflies. This parasitic disease presents mainly four clinical manifestations: visceral leishmaniasis (VL, also known as kala-azar), cutaneous leishmaniasis (CL), diffuse cutaneous leishmaniasis (DCL), and mucocutaneous leishmaniasis (MCL). According to the World Health Organization (WHO), 700,000 to 1 million new cases and 26,000 to 65,000 deaths occur yearly, constituting a major global public health problem [1,2].

The current treatment of leishmaniasis is still limited and is based mainly on the use of pentavalent antimonials. Unfortunately, these drugs have several disadvantages, including unsatisfactory efficacy, difficulty of administration, and numerous side effects due to their high toxicity. Furthermore, the development of drug-resistant parasites has been observed. Therefore, alternative treatments can be adopted in cases of pentavalent antimonials’ ineffectiveness. Among these are the use of amphotericin B and its liposomal formulations, pentamidine, paromomycin, azoles, and miltefosine. However, these drugs also present high toxicity, side effects, and high costs [3]. Given this scenario, searching for new effective and accessible antileishmanial agents has become a fundamental need.

Chalcones are an important group of compounds with a core of two aromatic rings linked by an α, β-unsaturated carbonyl chain, Ar-CO=CH-Ar, and they are found widely in nature. Additionally, they can be prepared readily via synthetic protocols [4]. Furthermore, it is in medicinal chemistry that chalcones are rightfully considered to have a privileged structure [5], since the chalcone core is present in a variety of important bioactive compounds with a broad spectrum of pharmacological activities, such as antimalarial, antibacterial, anticancer, anti-inflammatory, antiplasmodial, antioxidant, antifungal, antitubercular and antiviral agents [5,6]. Furthermore, recent reports have further indicated that chalcone derivatives (Figure 1) may also be a promising pharmacophoric group against diseases caused by protozoa, including leishmaniasis [7,8,9,10].

In the search for new biological agents, an important aspect is the identification of molecular targets to provide selective inhibition. For *Leishmania* spp., one such target is the enzyme arginase (ARG), responsible for L-arginine hydrolysis, which provides the necessary substrate for biosynthesis of polyamines required for growing cellular, replication, and infectivity of the parasite. [11]. Another important enzyme to target is trypanothione reductase (TR). This NADPH-dependent enzyme catalyzes trypanothione disulfide reduction in trypanothione dithiol, representing an essential system to combat oxidative stress caused by reactive oxygen species (ROS). A further reason for targeting TR is its absence in mammals [12].

We have synthesized and evaluated the antileishmanial potential of a series of nine new chalcone derivatives containing the 4,8-dimethoxynaphthalenyl moiety. The antipromastigote activity and cytotoxicity were assessed on murine macrophage cell lines (J774A.1). The potential of compounds was based on their IC_50_ and selectivity index (SI) values. In addition, the effect against amastigote forms, as well as such biological assays, as mitochondrial depolarization, reactive oxygen species generation, morphological, and ultrastructural analysis to understand the mechanisms of action in promastigote forms. Finally, we performed molecular docking on two targets of *L. amazonensis*, ARG and TR (Figure 2).

## 2. Results and Discussion

### 2.1. Chemistry

The general syntheses of compounds **4a**–**i** are shown in Figure 1 and involve successive reactions of 1,5-dihydroxynaphthalene, 1, (i) with iodomethane and potassium carbonate, (ii) acetic anhydride, BF_3_.etherate, and (iii) an aromatic aldehyde.

The final products **4a**–**i** were purified by flash chromatography and characterized by ^1^H, ^13^C, COSY, and HSQC NMR, IR, and HR mass spectrometry. In general, ^1^H NMR spectra showed two characteristic doublets between δ 7.38–7.00 and δ 7.11–6.90 with a coupling constant (J_Hα-Hβ_) around 16.0 Hz, indicating the formation of the (*E*)-diastereoisomer. ^13^C NMR spectra exhibited signals related to the C=O carbons between δ 199.23–197.01. In addition, the IR spectra showed two absorption bands around 3020 cm^−1^ and 1665 cm^−1^, which were assigned to the stretching vibrations of =C−H and carbonyl band (n C=O), respectively. The novel compounds showed similar IR and NMR spectroscopic signals, and the HRMS unequivocally confirmed their molecular formulas. 

#### Crystal Structure of Compound **4f** and comparisons with those of **4c** and **4i**

The atom arrangement and numbering scheme for compound **4f** are shown in Figure 3A. The molecule has a distinctive shape, between a “T” and a “Y”, with an interplanar angle between the naphthalene and phenyl rings of 51.08°, see Figure 3B. The nitro group is almost coplanar with its attached phenyl ring, and an *E* geometry about the exo C=C group is present. The bond lengths in the linker chain between the phenyl and naphthalene rings indicate a degree of delocalization, with bond lengths being intermediate between those expected for single and double bonds, as expected for chalcones, as reported earlier for compounds **4c** and **4i** [13].

Comparisons of the molecular structures of **4c** and **4i** [13] with those of **4f** indicate that different rotations about the C1–C11 bonds have resulted in different separations of the O4 methoxy oxygen and the O11 carbonyl oxygen, which provides the *anti-periplanar arrangements* in **4i** and **4f** and the *syn-periplanar* arrangement in **4c** within the bridging prop-2-en-1-one units, see Figure 4. The interplanar angles between the naphthalene and phenyl rings in **4c** and **4i** are 70.01(8) and 50.20(7)°, respectively, which also places the molecular structure of **4f** close to that of **4i.** Calculations [13] indicated that the stabilities of the *anti-* and *syn-periplanar* molecular arrangements are very similar, with intermolecular interactions being significant factors in the crystal structures.

Molecules of **4f** are linked in the crystalline state into a three-dimension array by a variety of interactions, including C-H---O hydrogen bonds and π---π, C-H---π, and N-O---π interactions. However, the methoxy oxygens (O4 and O9) are not involved in the intermolecular interactions, unlike the carbonyl oxygen (O11) and the nitro group oxygens (O17A and O17B).

### 2.2. Biological Evaluation

#### 2.2.1. Cytotoxicity and Antipromastigote Activity In Vitro of the 4,8-Dimethoxynaphthalenyl Chalcone Derivatives (**4a**–**i**)

The antileishmanial activities of compounds **4a**–**i** were initially evaluated in vitro with promastigote forms of *L. amazonensis*, related to American cutaneous leishmaniasis. Furthermore, as macrophages are the host cells critical for *Leishmania* spp., the cytotoxicities were tested against murine macrophages (J774A.1) by the method developed by Mosmann [14], based on mitochondrial oxidation. The antipromastigote activities varied widely, with IC_50_ values ranging from 3.3 ± 0.34 to 264.1 ± 0.12 μM. However, all compounds **4a**–**i** showed no toxicity to mammalian cells, showing CC_50_ values > 100 μM, see Table 1.

The results obtained from the preliminary analysis of the structure-activity relationship (SAR) indicate that the presence of electron-donating groups in the phenyl ring in **4a**–**b**, such as methoxy groups, leads to a decrease in the antileishmanial activity compared to the unsubstituted derivative **4h**, (IC_50_ = 26.1 ± 0.09 µM). However, this decrease was less for the meta-methoxyl derivative, **4b**, (IC_50_ = 67.1 ± 0.16 μM) than for the *para* derivative, **4a**, (IC_50_ = 264.1 ± 0.12 µM). The effects of different electron-withdrawing groups, such as Br (in **4i**), Cl (in **4c**, **4d**, and **4e**), and NO_2_ (in **4f** and **4g**), were also investigated. For compound **4i**, the presence of bromine led to a small increase in activity (IC_50_ = 24.3 ± 0.05 μM). Concerning the disubstituted chlorinated derivatives (**4c**–**e**), it was observed that the presence of at least one chlorine atom in an ortho position (**4d** and **4e**) promotes an increase in IC_50_ values (IC_50_ = 94.1 ± 0.06 and 37.7 ± 0.06 µM, respectively), which could be associated with the ortho effect on the biological activity [15]. This is more evident for the compound, which has two chlorine atoms at ortho positions, being the least active of the chloro-substituted compounds. Compound **4c,** without ortho-chlorine, showed the best activity of the chlorine-containing substituents and was even more active than the unsubstituted **4h** and the brominated **4i** derivatives. For the nitro-derivatives **4f**–**g**, good activities were found, with the *para* nitro derivative **4f** being the most active derivative of the series (IC_50_ = 3.3 ± 0.34 μM), only a slight reduction was found for the *meta*-nitro derivative (IC_50_ = 14.5 ± 0.09 μM).

This analysis shows that compound **4f** is the most promising in terms of activity (IC_50_ = 3.3 ± 0.34 µM) and selectivity index (SI = 112.6). In drug discovery for leishmaniases, the hit and lead criteria to elect a promising compound are an IC_50_ < 10 μM and a selectivity index ≥ 10 against parasitic protozoa [16]. Compound **4f** was selected for further in vitro assays, which included the ability to generate reactive oxygen species (ROS) and the depolarization of the mitochondrial membrane, as well as in morphological and ultrastructural analysis. All these were carried out to fully understand the mechanisms of eliminating promastigote forms and the anti-amastigote evaluation. Finally, a molecular docking study was performed on potential molecular targets (ARG and TR).

#### 2.2.2. Mechanism of Action in Promastigotes of *L. amazonensis*

The most effective response of macrophages against *Leishmania* spp. is mediated by producing toxic microbicide molecules, such as ROS and NO. These species trigger a well-established mechanism for eliminating the parasite, leading to infection resolution and parasite elimination without damaging host cells [17]. It is well known that high concentrations of ROS molecules can cause harmful effects in mitochondria, leading to a reduction in the membrane potential. Since this organelle is essential to ATP production, its dysfunctionality leads to parasite death [18].

A substantial rise in the intracellular ROS levels on treatment with compound **4f** was indeed found at both the tested concentrations using the method of conversion of the non-fluorescent probe (H_2_CFDA) to fluorescent (DCF), Figure 5A. 

As shown in Figure 5B, the tetramethylrhodamine ethyl ester (TMRE) cationic probe indicated that compound **4f** reduces the potential of the mitochondrial membrane (ΔΨm) compared to the two controls, leading to depolarization. However, no difference was found in the two tested concentrations.

#### 2.2.3. Morphological and Ultrastructural Changes in Promastigotes

SEM and TEM were used to detect the morphological and ultrastructural changes induced by the treatment with **4f** on the promastigote form (Figure 6 and Figure 7). SEM showed that untreated parasites (Figure 6A) and those treated with DMSO (0.01%) (Figure 6B) exhibited normal characteristics, compatible with an elongated body, flagellum proportional to body size, smooth and intact cell surface, and well-preserved morphology. However, parasites treated with **4f** at both 3.3 μM (Figure 6C–F) and 6.6 μM (Figure 6G–I) for 24 h showed morphological changes, such as a rounded shape and reduction of cell body size, cell surface roughness, and damage in the flagellum region.

TEM analysis indicated that both untreated parasites (Figure 7A) and those treated with DMSO (0.01%) (Figure 7B) had well-preserved structures of the nucleus, mitochondria, kinetoplast, flagellum region, and Golgi apparatus. However, treatment with increasing concentrations of **4f** 3.3 μM (Figure 7C–F) and 6.6 μM (Figure 7G–I) for 24 h showed an increase in autophagic vacuoles, lipid-storage bodies accumulation, mitochondrial swelling, and nuclear alterations.

#### 2.2.4. Evaluation of Antileishmanial Activity on *L. amazonensis*-Intracellular Amastigotes Forms

The antiamastigote effect of compound **4f** on *L. amazonensis*-infected macrophages was also investigated (Figure 8). Our results showed that 24 h after infection, the negative control, and DMSO vehicle groups were similar with 98.09 ± 0.44% and 96.97% ± 0.85% of infected macrophages, respectively. Treatment with **4f** at concentrations of 10, 25, 50, and 130 μM led to reductions to 23.81 ± 1.07%, 35.78 ± 1.59%, 42.15 ± 0.49%, and 63.04 ± 0.35%, respectively (Figure 8A). In addition, the number of amastigotes per macrophage was also reduced to 14.14 ± 0.04%, 36.20 ± 0.02%, 39.33 ± 0.03%, and 45. 27 ± 0.10% (Figure 8B) compared to controls, respectively. Notably, the IC_50_ of **4f** for intracellular amastigotes was 18.5 ± 1.19 µM. Analyzing the results on the intracellular stage, compound **4f** significantly reduced the percentage of infected macrophages and the number of amastigotes per macrophage at all tested concentrations (10–130 μM). The standard drug (AmB) resulted in a 95.25 ± 0.65% reduction in the percentage of infected macrophages and reduced by 89.55 ± 0.05% the number of amastigotes per macrophage. 

Our results are similar to those from other studies on amastigote forms of *L. braziliensis* [19], *L. amazonensis*, *L. braziliensis*, and *L. peruviana* [20], *L. donovani* [21], confirming that chalcones, in general, possess good antileishmanial activity.

#### 2.2.5. In Silico Study to Predict Pharmacokinetic and Toxicity Parameters ADME Prediction 

We also conducted in silico analysis of the physicochemical and pharmacokinetic properties to predict the behavior of compound **4f** in living organisms (Table 2).

The analysis of physicochemical properties indicates that derivative **4f** does not violate Lipinski’s rule and thus demonstrates that **4f** could have good bioavailability after being administered orally [22] and meets the parameters proposed by Veber, indicating good permeability through membranes [23]. Furthermore, the value of Log *S* [24] showed that compound **4f** is moderately soluble in water.

Regarding the pharmacokinetic properties, our study has shown that compound **4f** has high gastrointestinal absorption and cannot permeate the blood-brain barrier (BBB), whose function is in the maintenance of homeostasis in the central nervous system [25]. P-glycoprotein (P-gp) is an efflux transporter that functions as a pump, sending xenobiotics out of the cell, playing a significant role in the pharmacokinetic and pharmacodynamic properties of drugs, such as reducing bioavailability [26]. This compound does not act as a substrate for P-gp. Therefore, no significant changes in absorption, distribution, and elimination of compound **4f** are expected.

Cytochrome P450 (CYP) enzymes are essential in metabolizing various xenobiotics, including drugs [27]. Herein, we used a forecast model with five CYP450 isoforms of CYP1, CYP2, and CYP3 families (CYP1A2, CYP2C19, CYP2C9, CYP2D6, and CYP3A4). Compound **4f** showed the ability to inhibit three isoforms, CYP2C19, CYP2C9, and CYP3A4. CYP2C19 is primarily responsible for the metabolism of drugs such as proton pump inhibitors and antidepressants [28], so it is unlikely that important drug interactions will occur in patients with leishmaniasis. However, CYP3A4 is the most abundant hepatic and intestinal phase I enzyme, responsible for the metabolism of more than 50% of drugs [29], and CYP2C9 is responsible for approximately 15% of drug metabolism, including non-steroidal anti-inflammatory [30], used in the treatment of cutaneous leishmaniasis. The inhibition of these isoforms can decrease the efficacy and increase the side effects and toxicity of co-administered drugs, indicating the need for further studies to verify the interaction with drugs metabolized by these CYP’s enzymes.

In addition, we also verified if the compound **4f** could act as a possible substrate for CYPs. This prediction is useful because if a substance behaves as a substrate of CYPs, it can lead to a low oral bioavailability caused by a pre-systemic metabolism. Our predictions showed that compound **4f** could be a substrate of CYP1A2, CYP2C9, CYP2D6, and CYP3A4 isoforms. In addition, through analyzing sites of metabolism (SoMs), we predicted the positions of metabolically labile atoms in the substrate, where metabolic reactions can be initiated. The study showed that the carbons of the methoxy groups, at 4 and 8 positions of the naphthalene ring, are susceptible to the action of CYPs since these enzymes are known to perform hydroxylation of α-carbons to heteroatoms that will result in O-demethylation [31]. 

### 2.3. Molecular Docking

ARG [32] and TR [33] are two important molecular targets of *L. amazonensis* since they play a pivotal role in the growth and survival of the parasite. Therefore, we decided to investigate the ligand-protein interactions between chalcone **4f** and these targets through a molecular docking study.

Owing to the lack of experimental 3D structures of ARG and TR of *L. amazonensis*, our 3D models were built by homologic modeling. Therefore, the models were built with consideration of the structures of the ARG in *L. mexicana* (95.4% identity) and TR in *C. fasciculata* (76.8% identity).

Consensus scoring is a useful validation method as it improves the scoring’s fitting performance and improves the prediction of bound conformations and poses [34]. We applied it in our study to predict and visualize the most favorable interactions between **4f** and active sites, considering the consensus between at least two scoring functions (RMSD < 2.0). The analysis of the consensus generated from the best pose obtained by different scoring functions used in this study presented an RMSD of 0.7463 between the Lamarckian genetic algorithm and genetic algorithm for the ARG enzyme and an RMSD of 1.762 between the same functions for the TR enzyme.

ARG is a homotrimer (Figure 9A), with each subunit containing a binuclear Mn (II) center critical for catalytic activity. In our docking study, only the active site of chain A was considered to indicate protein-ligand interactions. The docking analysis for the monomeric model of ARG indicated that compound **4f** is hydrogen bonded to the residues Asn143, His139, and His114. These last two are pivotal since His139 is conserved in the ARG family and is involved in coordination with the binuclear manganese cluster in the active site [35], and His114 directly participates in the catalytic mechanism of this enzyme [36]. The hydrogen bonds between **4f** and the protein contribute significantly to the stability of the protein-ligand complex. Furthermore, our studies also demonstrated that the nitro group is involved in coordination with the Mn (II) centers (Figure 9B). This is clearly of importance not only to the complex stability, but also to the inhibition of the enzymatic activity involving the Mn (II) centers [37]. It is also important to note that compound **4f** exhibited hydrophobic interactions with residues Asp141, Ser150, His154, Gly155, and Glu197 (Figure 9C). 

The active TR structure is homodimeric (Figure 10A), with each monomer composed of three domains—a domain that contains an active binding site for T(S)_2_ and allosteric binding domains for FAD and NADPH binding [38]. We defined the docking site only as the substrate-binding site, not including the allosteric regions for our molecular docking study. Since FAD is strongly linked to TR, this site would not be available for inhibitors in vitro [39]. 

TR’s active site is a large crevice located at the interface of chain A and chain B. In this interface, there is a channel that connects the two active sites [40]. Given this structural characteristic and analyzing the complex between compound **4f** and the TR enzyme, we can observe that the nitro group is partially inserted in this channel, forming two hydrogen bonds with residues Lys530 and Thr534. On the other hand, the methoxy group of 4,8-dimethoxynaphthalenyl lies outside the channel and interacts with the side chain of Thr390 (Figure 10B). In addition to the hydrogen bond interactions, the compound also interacts hydrophobically with residues Leu392, Met393, Phe389, Leu531, and Ile575 (Figure 10C). These interactions observed in the protein-ligand complex may be related to enzyme inhibition, leading to the observed increase in ROS levels since TR plays a unique role in trypanothione-based redox metabolism and oxidative defense.

The binding energy calculated in the docking study for the **4f**-ARG enzyme interaction (−11.94 kcal/mol) indicates greater stability than that for the **4f**-TR enzyme (−7.61 kcal/mol), i.e., these results suggest that **4f** interacts more efficiently with critical residues at the active site of ARG, suggesting that derivate **4f** may be considered a possible pharmacophoric group to inhibit this target. Consequently, these results suggest that this compound should be indicated for further antileishmanial studies.

## 3. Materials and Methods

### 3.1. General Information

Solvents and reagents were purified following the literature procedures [41]. Melting points were determined on a Fisatom 431 hot plate apparatus and were uncorrected. HRMS spectra of the title compounds in methanol (MS degree) solutions (1 mg/mL) with acetic acid 0.1% were obtained on a high-resolution quadrupole-TOF electrospray mass spectroscopy (Bruker, model Impact HD). Infrared spectra (400–4000 cm^−1^) were recorded as KBr discs on a Nicolet 6700. NMR spectra were obtained on a Bruker spectrometer Model Avance III operating at 500 MHz for ^1^H and 125 MHz for ^13^C or a Bruker spectrometer Model AVHD operating at 400 MHz for ^1^H and 100 MHz for ^13^C. NMR resonances were registered using CDCl_3_ (Merck, Darmstadt, Germany) as solvent and TMS as the internal standard in CDCl_3_. Chemical shifts (δ in ppm) were referenced to the TMS signal in CDCl_3_ at δ 0.00. The splitting of proton resonances in the reported ^1^H NMR spectra is defined as singlet (s), doublet (d), triplet (t), quadruplet (qua), quintuplet (qui), and complex pattern (m). Coupling constants (*J*) are reported in Hz (Hertz). Iodomethane, potassium carbonate, acetic anhydride, boron trifluoride diethyl etherate and the aromatic aldehydes (3-methoxybenzaldehyde, 4-methoxybenzaldehyde, 3, 4-dichlorobenzaldehyde, 2, 6-dichlorobenzaldehyde, 2, 4-dichlorobenzaldehyde, 3-nitrobenzaldehyde, 4-nitrobenzaldehyde, 4-bromobenzaldehyde) were purchased from Sigma Aldrich (São Paulo, Brazil). Analytical thin-layer chromatography was carried out with E. Merck silica gel 60 F254 coated 0.25 plates and visualized with a long- and short-wavelength UV lamp. Flash column chromatography was conducted over Merck silica gel 60 (0.040–0.063 mm). Ethyl acetate and n-hexane were used as eluent.

### 3.2. Chemistry

#### 3.2.1. Synthesis of 1,5-Dimethoxynaphthalene **(2)**

A glass pressure vessel was charged with 1,5-dihydroxynaphtalene **(1)** (5.05 g, 31.5 mmol), K_2_CO_3_ (4.76 g, 34.3 mmol), iodomethane (8.94 g, 63.0 mmol) and acetonitrile (120 mL). The reaction vessel was sealed with a PTFE bushing, the solution was heated to 80 °C with magnetic stirring for 24 h, cooled to room temperature, acetonitrile was removed under a vacuum, and then water (200 mL) was added. The aqueous phase was extracted with chloroform (3 × 50 mL). The organic phase was dried over sodium sulfate and concentrated under a vacuum. Product **(2)** was obtained in an 86% yield and used without further purification. 

***1,5-dimethoxynaphtalene (2):*** Yield: 86%. ^1^H NMR (CDCl_3_, 400 MHz) δ: 7.85 (2H, d, H-4, H-8, *J* = 8.44 Hz), 7.39 (2H, t, H-3, H-7, *J* = 8.00 Hz), 6.86 (2H, d, H-2, H-6, *J* = 7.64 Hz), 4.00 (6H, s, -OCH_3_). ^13^C NMR (CDCl_3_, 400 MHz) δ: 155.27, 126.63, 125.17, 114.21, 104.56, 55.55.

#### 3.2.2. Synthesis of 1-(1,5-Dimethoxynaphthalen-8-yl)ethenone **(3)**

To a solution of 1,5-dimethoxynaphthalene **(2)** (2.5 g, 13.3 mmol) in freshly distilled acetic anhydride (20 mL) at 0 °C under argon was added dropwise with stirring, boron trifluoride etherate (5.2 mL, 42.2 mmol) via a syringe. The mixture was stirred to 0 °C for 40 min, 10% NaOH aqueous solution (100 mL) was added, and the resulting solution was extracted with ethyl acetate (3 × 20 mL). The organic phase was washed with brine, dried over Na_2_SO_4,_ and concentrated under a vacuum to afford the crude product, which was purified by flash chromatography (hexane/EtOAc 8:2). The purified product **(3)** was obtained in a 55% yield.

***1-(1,5-dimethoxynaphthalen-8-yl)ethenone* (3)**: Yield: 55%. ^1^H NMR (CDCl_3_, 500 MHz) δ: 7.90 (1H, d, H-7, *J* = 8.40 Hz), 7.43 (1H, t, H-3, *J* = 8.25 Hz), 7.20 (2H, d, H-4, *J* = 7.85 Hz), 6.91 (4H, t, H-6, *J* = 7.70 Hz), 6.79 (2H, t, H-2, *J* = 7.90 Hz), 3.99 (3H, s, OCH_3_), 3.91 (3H, s, OCH_3_) 2.46 (3H, s, -CH_3_). ^13^C NMR (CDCl_3_, 500 MHz) δ: 156.10, 154.50, 131.98, 126.82, 126.07, 123.56, 122.70, 114.97, 106.67, 103.50, 55.70, 55.58, 31.99.

#### 3.2.3. General Procedure for the Synthesis of 4,8-Dimethoxynaphthalenyl Chalcone Derivatives **(4a**–**i)**

The 4,8-dimethoxynaphthalenyl chalcone derivatives **(4a**–**i)** were prepared through the reaction between the ketone **(3)** and the appropriately substituted benzaldehyde. 

In a round bottom flask, ketone **(3)** (0.24 g, 1.06 mmol), KOH (0.17 g, 4.24 mmol), the aldehyde (1.2 mmol), and a mixture of methanol/water (40 mL, ratio 8:2) were added. The solution was heated to 90 °C, with magnetic stirring overnight. Then, methanol was removed under a vacuum, HCl aqueous solution (15 mL, 6 M) was added, and the resulting solution was extracted with chloroform (3 × 10 mL). Next, the organic phase was washed with brine, dried over Na_2_SO_4,_ and concentrated under a vacuum. Finally, the crude product was purified by flash chromatography (using 20% ethyl acetate/hexane as eluent).

***(E)-1-(4,8-dimethoxynaphthalen-1-yl)-3-(4-methoxyphenyl)prop-2-en-1-one (4a):*** Yield: 52%, brown solid, melting point 97–102 °C [42]. ^1^H NMR (CDCl_3_, 400 MHz) δ: 7.93 (1H, dd, H-4, *J* = 8.48 and 0.8 Hz), 7.45–7.38 (3H, m, H-3″, H6″ e H2″), 7.29 (1H, d, H-7, *J* = 7.9 Hz), 7.07 (1H, d, H-3′, *J* = 16.1 Hz), 6.93 (1H, d, H-2′, *J* = 16.1 Hz), 6.91–6.84 (4H, m, H-2, H6, H3″, H5″), 4.02 (3H, s, -OCH_3_), 3,80 (6H, s, -OCH_3_). ^13^C NMR (CDCl_3_, 400 MHz) δ: 198.65, 161.22, 156.14, 154.96, 142.83, 129.75, 129.59, 127.68,127.18,126.90, 126.07, 125.49, 123.79, 114.81, 114.30, 113.31, 112.78, 106.95, 103.60, 55.74, 55.58, 55.37. IR n, cm^−1^: 3010 (=C-H), 1647 (C=O), 1601 (C=C). HRMS-ESI: Calcd. for C_22_H_20_O_4_ [M + H]^+^ 349.1440, found 349.1431.

***(E)-1-(4,8-dimethoxynaphthalen-1-yl)-3-(3-methoxyphenyl)prop-2-en-1-one (4b):*** Yield: 50%, red solid, melting point 81–86 °C. ^1^H NMR (CDCl_3_, 400 MHz) δ: 7.97 (1H, d, H-4, *J* = 8.5 Hz), 7.48 (1H, t, H-3, *J* = 7.9 Hz), 7.34 (1H, d, H-7, *J* = 7.9 Hz), 7.29 (1H, d, H-5″, *J* = 5.9 Hz), 7.11 (1H, d, H-3′, *J* = 16.1 Hz), 7.08 (1H, d, H-6″), 7.03 (1H, d, H-2′, *J* = 16.1 HZ), 6.99–6.89 (4H, m, H-2, H-6, H-2″, H-4″), 4.07 (3H, s, -OCH_3_), 3.84 (3H, s, -OCH_3_), 3.82 (3H, s, -OCH_3_). ^13^C NMR (CDCl_3_, 400 MHz) δ: 198.43, 159.88, 156.31, 154.85, 142.49, 136.45, 129.80, 129.48, 126.89, 126.13, 125.60, 123.76, 120.79, 119.11, 115.93, 114.86, 112.90, 106.98, 103.61, 55.76, 55.56, 55.29. IR n, cm^−1^: 3027 (=C-H), 1656 (C=O), 1627 (C=C). HRMS-ESI: Calcd. for C_22_H_20_O_4_ [M + H]^+^ 349.1440, found 349.1423.

***(E)-3-(3,4-dichlorophenyl)-1-(4,8-dimethoxynaphthalen-1-yl)prop-2-en-1-one******(4c)***: Yield: 15%, grey solid, melting point 109–113 °C [13]. ^1^H-NMR (CDCl_3_, 500 MHz) δ: 7.92 (1H, d, H-4, *J* = 8.5 Hz), 7.48 (1H, s, H-2″), 7.43 (1H, t, H-3, *J* = 8.5 Hz), 7.38 (1H, d, H-5″, *J* = 8.4 Hz), 7.29 (1H, d, H-7, *J* = 7.9 Hz), 7.25 (1H, d, H-6″, *J* = 8.4 Hz), 7.00 (1H, d, H-3′, *J* = 16.1 Hz), 6.96 (1H, d, H-2′, *J* = 16.1 Hz), 6.89 (1H, d, H-2, *J* = 7.7 Hz), 6,84 (1H, d, H-6, *J* = 7.9 Hz), 4.02 (3H, s, -OCH_3_), 3.79 (3H, s, -OCH_3_). ^13^C NMR (CDCl_3_, 500 MHz) δ: 197.39, 156.55, 154.70, 138.90, 135.24, 133.68, 133.14, 130.79, 130.65, 129.60, 129.18, 126.91, 126.19, 126.01, 125.77, 123.67, 115.06, 107.13, 103.62, 55.77, 55.55. IR n, cm^−1^: 3005 (=C-H), 1647 (C=O), 1630 (C=C), 1589 (C=C). HRMS-ESI: Calcd. for C_21_H_16_Cl_2_O_3_ [M + H]^+^ 387.0555, found 387.0532.

***(E)-3-(2,6-dichlorophenyl)-1-(4,8-dimethoxynaphthalen-1-yl)prop-2-en-1-one (4d)*****:** Yield: 20%, grey solid, melting point 144–145 °C. ^1^H NMR (CDCl_3_, 400 MHz) δ: 7.91 (1H, d, H-4, *J* = 8.4 Hz), 7.43 (1H, t, H-3, *J* = 7.80 Hz), 7.37 (1H, d, H-7, *J* = 7.8 Hz), 7.28 (2H, t, H-3″e H-5″, *J* = 8.05 Hz), 7.11 (1H, t, H-2′, *J* = 15.5 Hz), 7.10 (1H, t, H-4″, *J* = 7.6 Hz), 7.06 (1H, d, H-3′, *J* = 15.5 Hz), 6.89 (1H, d, H-2, *J* = 8.4 Hz), 6.87 (1H, d, H-6, *J* = 8.4 Hz), 4.02 (3H, s, -OCH_3_), 3.85 (3H, s, -OCH_3_). ^13^C NMR (CDCl_3_, 500 MHz) δ:198.04, 156.47, 154.64, 137.11, 135.79, 134.66, 132.84, 129.45, 128.85, 128.62, 127.52, 126.78, 126.18, 125.87, 125.71, 123.72, 114.72, 106.54, 103.64, 55.74, 55.37. IR n, cm^−1^: 3002 (=C-H), 1650 (C=O), 1589 (C=C). HRMS-ESI: Calcd. for C_21_H_16_Cl_2_O_3_ [M + H]^+^ 387.0555, found 387.0463.

***(E)-3-(2,4-dichlorophenyl)-1-(4,8-dimethoxynaphthalen-1-yl)prop-2-en-1-one (4e):*** Yield: 75%, brown solid, melting point 130–134 °C. ^1^H NMR (CDCl_3_, 400 MHz) δ: 7.93 (1H, d, H-4, *J* = 8.4 Hz), 7.52 (1H, d, H-6″, *J* = 8.4 Hz), 7.48 (1H, d, H-3′, *J* = 15.8 Hz), 7.44 (1H, t, H-3, *J* = 8.2 Hz), 7.35 (1H, d, H-3″, *J* = 1.5 Hz), 7.34 (1H, d, H-7, *J* = 7.8 Hz), 7.22 (1H, dd, H-5″, *J* = 8.5 Hz and 1.5 Hz), 6.95 (1H, d, H-2′, *J* = 16.1 Hz), 6.90 (1H, d, H-2, *J* = 7.7 Hz), 6.86 (1H, d, H-6, *J* = 7.9 Hz), 4.03 (3H, s, -OCH_3_), 3.81 (3H, s, -OCH_3_). ^13^C NMR (CDCl_3_, 400 MHz) δ:156.57, 154.66, 136.29, 135.80, 135.38, 132.18, 131.77, 129.86, 129.23, 128.29, 127.47, 126.88, 126.16, 125.88, 123.70, 114.99, 107.08, 103.23, 103.60, 55.75, 55.45. IR n, cm^−1^: 3075(C-H Ar), 1661(C=O), 1605(C=C). HRMS-ESI: Calcd. for C_21_H_16_Cl_2_O_3_ [M + H]^+^ 387.0555, found 387.0487.

***(E)-1-(4,8-dimethoxynaphthalen-1-yl)-3-(4-nitrophenyl)prop-2-en-1-one (4f)******:*** Yield: 85%, orange solid, melting point 192–195 °C. ^1^H NMR (CDCl_3_, 400 MHz) δ: 8.18 (2H, d, H-3″ e H-5″, *J* = 8.8 Hz), 7.95 (1H, d, H-4″, *J* = 8.5 Hz), 7.58 (2H, d, H-2″, H-6″ *J* = 8.8 Hz), 7.46 (1H, t, H-3, *J* = 7.9 Hz), 7.34 (1H, d, H-7, *J* = 7.9 Hz), 7.17 (1H, d, H-3′, *J* = 16.1 Hz), 7.11 (1H, d, H-2′, *J* = 16.1 Hz), 6.92 (1H, d, H-2, *J* = 7.7 Hz), 6.87 (1H, d, H-6, *J* = 7.9 Hz), 4,05 (3H, s, -OCH_3_), 3,81 (3H, s, -OCH_3_). ^13^C NMR (CDCl_3_, 400 MHz) δ: 197.01, 156.74, 154.60, 148.21, 141.46, 138.13, 132.64, 129.20, 129.00, 128.51, 126.92, 126.28, 126.00, 124.07, 123.62, 122.35, 115.15, 107.20, 103.63, 55.80, 55.53. IR n, cm^−1^: 3020 (=C-H), 1667 (C=O), 1593 (C=C), 1575 (N=O), 850 (N=O). HRMS-ESI: Calcd. for C_21_H_17_NO_5_ [M + H]^+^ 364.1185, found 364.1142.

***(E)-1-(4,8-dimethoxynaphthalen-1-yl)-3-(3-nitrophenyl)prop-2-en-1-one (4g):*** Yield: 73%, yellow solid, melting point 77–79 °C. ^1^H NMR (CDCl_3_, 500 MHz) δ: 8.26 (1H, dd, H-2″, *J*1 = 1.6 Hz, *J*2 = 1.5 Hz), 8.17 (1H, dd, H-4″, *J*1 = 8.2 Hz, *J*2 = 1.4 Hz), 7.95 (1H, d, H-4, *J* = 8.4 Hz), 7.76 (1H, d, H-6″, *J* = 7.8 Hz), 7.52 (1H, t, H-5″, *J* = 7.9 Hz), 7.46 (1H, t, H-3, *J* = 8.0 Hz), 7.33 (1H, d, H-7, *J* = 7.9 Hz), 7.16 (1H, d, H-3′, *J* = 16.1 Hz), 7.11 (1H, d, H-2′, *J* = 16.1 Hz), 6.92 (1H, d, H-2, *J* = 7.7 Hz), 6.87 (1H, d, H-6, *J* = 7.9 Hz), 4.05 (3H, s, -OCH_3_), 3.81 (3H, s, -OCH_3_). ^13^C NMR (CDCl_3_, 500 MHz) δ: 197.21, 156.68, 154.62, 148.70, 138.51, 136.98, 133.47, 131.69, 129.83, 129.00, 126.93, 126.27, 125.87, 124.08, 123.63, 122.43, 115.13, 107.19, 103.61, 55.80, 55.58. IR n, cm^−1^: 3039 (=C-H), 1669 (C=O), 1612 (C=C), 1577 (N=O). HRMS-ESI: Calcd. for C_21_H_17_NO_5_ [M + H]^+^ 364.1185, found 364.1203.

***(E)-1-(4,8-dimethoxynaphthalen-1-yl)-3-phenylprop-2-en-1-one (4h):*** Yield: 71%, brown solid, melting point 141–144 °C. ^1^H NMR (CDCl_3_, 400 MHz) δ: 7.93 (1H, d, H-4, *J* = 8.5 Hz), 7.46–7.40 (3H, m, H-3, H2″, H-6″), 7.36–7.30 (4H, m, H-7, H-3″, H-4″, H-5″), 7.11 (1H, d, H-3′, *J* = 16.1 Hz), 7.04 (1H, d, H-2′, *J* = 16.1 Hz), 6.89 (1H, d, H-2, *J* = 7.7 Hz), 6.86 (1H, d, H-6, *J* = 7.9 Hz), 4.04 (3H, s, -OCH_3_), 3.80 (3H, s, -OCH_3_). ^13^C NMR (CDCl_3_, 400 MHz) δ: 198.59, 156.30, 154.85, 142.69, 135.04, 129.98, 129.64, 129.36, 129.21, 128.83, 128.11, 127.71, 126.88, 126.13, 125.61, 123.76, 114.86, 106.97, 103.60, 55.76, 55.55. IR n, cm^−1^: 3060 (=C-H), 1651 (C=O), 1608 (C=C). HRMS-ESI: Calcd. for C_21_H_18_O_3_ [M + H]^+^ 319.1334, found 319.1352.

***(E)-3-(4-bromophenyl)-1-(4,8-dimethoxynaphthalen-1-yl)prop-2-en-1-one (4i):*** Yield: 52%, yellow solid, melting point 162–164 °C [13]. ^1^H NMR (CDCl_3_, 500 MHz) δ: 7.93 (1H, d, H-4, *J* = 8.5), 7.47–7.43 (3H, m, H-3, H-3″ e H-5″), 7.31–7.25 (3H, m, H-7. H-2″ e H-6″), 7.05 (1H, d, H-3′, *J* = 16.1 Hz), 7.00 (1H, d, H-2′, *J* = 16.1 Hz), 6.90 (1H, d, H-2, *J* = 7.6 Hz), 6.86 (1H, d, H-6, *J* = 7.9 Hz), 4.04 (3H, s, -OCH_3_), 3.80 (3H, s, -OCH_3_). ^13^C NMR (CDCl_3_, 500 MHz) δ: 197.88, 156.42, 154.79, 140.66, 134.04, 132.05, 129.72, 129.42, 129.34, 126.91, 126.14, 125.68, 124.08, 123.72, 114.96, 107.04, 104.19, 103.60, 55.76, 55.54. IR n, cm^−1^: 3009 (C-H Ar), 1667 (C=O), 1606 (C=C), 606 (C-Br). HRMS-ESI: Calcd. for C_21_H_17_BrO_3_ [M + H]^+^ 397.0439, found 397.0465.

#### 3.2.4. Experimental X-ray Structure Determination

Data for compound **4f** were obtained with Mo-Kα radiation at 100 K using an *XtaLAB AFC11* (*RCD3): quarter*-*chi single* diffractometer at the NCS crystallographic service based at the University of Southampton, England. Data collection, data reduction, and unit cell refinement were achieved using CrysAlisPro 1.171.40.39a [43]. Correction for absorption was achieved with CrysAlisPro 1.171.40.39a [43], and empirical absorption correction using spherical harmonics, implemented in the SCALE3 ABSPACK scaling algorithm. The programs ORTEP-3 for Windows [44] and MERCURY [45] were used to prepare the figures. SHELXL97 [46] and PLATON [47] were used to calculate molecular geometry. The structures were solved by direct methods using SHELXS-97 [48] and fully refined using the program SHELXL-97 [46]. Finally, all hydrogen atoms were placed in calculated positions.

### 3.3. Biological Assays

#### 3.3.1. *Leishmania (Leishmania)* Amazonensis Maintenance

The promastigote forms of *L. amazonensis* (MHOM/BR/1989/166MJO) were maintained in culture medium 199 (GIBCO, Invitrogen, New York, NY, USA) pH 7.18–7.22 supplemented with 10% fetal bovine serum (FBS) (GIBCO, Invitrogen, New York, NY, USA), 1M HEPES buffer, 1% human urine, 1% l-glutamine, 10 µg/mL streptomycin and 10 U/mL penicillin (GIBCO, Invitrogen, NY, USA) and 10% sodium bicarbonate. The parasite culture was maintained in a B.O.D. incubator at 24 °C in 25 cm^2^ culture flasks. In all experiments, promastigote forms in the stationary growth phase were used (5-day culture).

#### 3.3.2. Effect of the Compounds on *L. amazonensis* Promastigotes

The activities were evaluated by the 3-(4,5-dimethylthiazol-2-yl)-2,5-diphenyltetrazolium bromide (MTT) assay [14]. *L. amazonensis* promastigote forms (10^6^ cells/mL) were placed in 96-well plates and treated with the chalcone derivatives (**4a**–**i**) at different concentrations (10, 25, 50, and 130 µM) and maintained in a B.O.D incubator for 24 h at 24 °C. A solution of MTT (10 μL of a 5 mg/mL stock) (Sigma-Aldrich, St. Louis, MO, USA) was added to each well, and after 4 h, the plates were evaluated at λ = 550 nm using a spectrophotometer (Thermo Fisher Scientific, Multiskan GO, Waltham, MA, USA). *L. amazonensis* promastigotes maintained in culture M199 medium without treatment or with 0.01% DMSO (Sigma-Aldrich, St. Louis, MO, USA) served as negative and vehicle controls. 1 μM amphotericin B (AmB) (Sigma-Aldrich, St. Louis, MO, USA) was used as a positive control. The results of antipromastigote activities were expressed as half-maximal inhibitory concentrations for 50% of parasites (IC_50_) in μM. The IC_50_ values were calculated as the average of three independent experiments performed in duplicates.

#### 3.3.3. Evaluation of Cytotoxicity on Murine Macrophages

The cytotoxic effect of compounds **4a**–**i** on murine macrophage cell lines J774A.1 (TIB-67, ATCC, Manassas, VA, USA) was assessed using the MTT assay (3-(4,5-dimethylthiazol-2-yl)-2,5-diphenyltetrazolium bromide) (Sigma-Aldrich, St. Louis, MO, USA), [14]. Macrophages were seeded in 96-well plates at a density of 10^4^ cells/well and were cultured with **4a**–**i** at different concentrations (10, 25, 50, and 130 µM) for 24 h (37 °C, 5% CO_2_). The cells were washed, MTT (5 mg/mL) added, and the cells were incubated for 4 h. The MTT product was diluted with 100 μL of DMSO (Sigma-Aldrich, St. Louis, MO, USA) and analyzed at 550 nm in a spectrophotometer (Thermo Fisher Scientific, Multiskan GO, Waltham, MA, USA) at 550 nm. 0.5% H_2_O_2_ (Merck, Darmstadt, Germany) was used as a positive control. The results were expressed as the percentage of viable cells compared to the control group. The cytotoxic concentration for 50% of the cells (CC_50_) was calculated by nonlinear regression to the dose-response curve using GraphPad Prism statistical software (GraphPad Software, Inc., San Diego, CA, USA, 500.288). 

#### 3.3.4. Selectivity Index (SI)

The half-maximal inhibitory concentration for 50% of parasites (IC_50_) was determined on *L. amazonensis* promastigotes treated with the nine compounds, and the cytotoxic concentration that causes the death of 50% of cells (CC_50_) on murine macrophages (J774A.1). IC_50_ and CC_50_ were calculated by nonlinear regression using GraphPad Prism statistical software (GraphPad Software, Inc., USA, 500.288). The tested compounds’ selectivity index was expressed as SI = CC_50_ on peritoneal macrophages/IC_50_ on promastigotes forms.

#### 3.3.5. Determination of Mitochondrial Membrane Potential (ΔΨm)

To evaluate if the most promising chalcone (**4f**) alters the mitochondrial membrane potential, we conducted an assay using a tetramethylrhodamine ethyl ester (TMRE) staining (Sigma-Aldrich, St. Louis, MO, USA). For this purpose, promastigotes (10^6^ cells/mL) were treated with 3.3 μM (IC_50_) and 6.6 μM (2× IC_50_) for 24 h and incubated with 25 nM of TMRE for 30 min at 24 °C and analyzed using a fluorescence microplate reader (Victor X3, PerkinElmer, Turku, Finland), at 480/580 nm. Carbonyl cyanide m-chlorophenylhydrazone (100 μM) (CCCP) (Sigma-Aldrich, St. Louis, MO, USA) was used as a positive control, and the vehicle (DMSO 0.01%) was used as a negative control.

#### 3.3.6. Reactive Oxygen Species (ROS) Generation in *L. amazonensis* Promastigotes

To evaluate the ROS generation in promastigotes forms of *L. amazonensis*, parasites (10^6^ cells/mL) were treated with the most promising chalcone (**4f**) at 3.3 μM (IC_50_) and 6.6 μM (2× IC_50_) for 24 h and incubated with 10 μM of a permeant probe, diacetate 2’,7’-dichlorofluorescein (H_2_DCFDA) (Sigma-Aldrich, St. Louis, MO, USA), diluted in DMSO in the dark for 45 min, 24 °C, with conversion to the highly fluorescent 2′,7′ -dichlorofluorescein (DCF). As a positive control, hydrogen peroxide (0.4% H_2_O_2_) was used, and the vehicle (DMSO 0.01%) was used for a negative control. ROS was measured as an increase in fluorescence caused by converting the non-fluorescent dye to the fluorescent 2.7-dichlorofluorescein, with an excitation wavelength of 488 nm and emission of 530 nm in a fluorescence microplate reader (Victor X3, PerkinElmer, Turku, Finland).

#### 3.3.7. Scanning Electron Microscopy (SEM) and Transmission Electron Microscopy (TEM)

SEM was performed to analyze morphological changes in the cell surface topography. Promastigotes (10^6^ parasites/mL) were treated with 3.3 μM (IC_50_) and 6.6 μM (2× IC_50_) of **4f** for 24 h at 25 °C. After the treatment, the parasites were fixed in 2.5% glutaraldehyde (Merck, Darmstadt, Germany) in 0.1 M sodium cacodylate buffer (Sigma-Aldrich, St. Louis, MO, USA) and adhered to coverslips covered with poly-L-lysine (Sigma-Aldrich, St. Louis, MO, USA) for 60 min. Parasites were dehydrated with increasing ethanol (Synth, Diadema, Brazil) concentrations (30–100%), submitted to critical point drying (Baltec SCD-030), metalized with gold, and visualized on a high-resolution double beam electron microscope FEI SCIOS.

To evaluate the ultrastructural changes by TEM, promastigotes were treated and fixed as described above. After the samples were post-fixed with 1% OsO_4_ (Sigma-Aldrich, St. Louis, MO, USA), 0.8% potassium ferrocyanide (Sigma-Aldrich, St. Louis, MO, USA), and 10.0 mM CaCl_2_ (Synth, Diadema, Brazil) in 0.1 M sodium cacodylate buffer for 1 h at room temperature and protected from light. Samples were washed in 0.1 M sodium cacodylate buffer, dehydrated in increased concentration of acetone (Synth, Diadema, Brazil) (50–100%), including in EPON resin, and polymerized at 60 °C for 72 h. Ultrathin sections were obtained in ultramicrotome (RMC Boeckler, Tucson, AZ, USA), deposited on a copper grid, and contrasted with 5% uranyl acetate (Sigma-Aldrich, St. Louis, MO, USA) and 2% lead citrate (Sigma-Aldrich, St. Louis, MO, USA) for 20 and 10 min, respectively. The analysis was performed using a transmission electron microscope JEOL JEM 1400.

#### 3.3.8. Antiamastigote Assay

Murine macrophages J774A.1 (5 × 10^5^ cell/mL) were cultivated in 24-well plates. After infection, the non-internalized promastigotes were removed by washing with PBS, and the adherent macrophages infected were treated with the most promising chalcone (**4f**) in non-cytotoxic concentrations (10, 25, 50, and 130 μM) for 24 h (37 °C, 5% CO_2_). RPMI 1640 medium, DMSO (0.01%), and AmB (1 μM) were used as a negative control, vehicle, and positive control, respectively. Then, cells were stained with Leishman (Laborclin, Pinhais, Brazil), and 20 fields were analyzed by increased immersion (1000× magnification) using an optical microscope (Olympus BX41, Olympus Optical Co., Ltd., Tokyo, Japan) to determine the % of infected macrophages and the number of amastigotes/cells.

#### 3.3.9. Statistical Analysis

Data were expressed as mean ± standard error of the mean (SEM). Three independent experiments were performed, each with duplicate datasets. Data were analyzed using the GraphPad Prism statistical software (GraphPad Software, Inc., USA, 500.288). Significant differences between the groups were determined by a *t*-test and one-way ANOVA, followed by Tukey’s test for multiple comparisons. Differences were considered statistically significant when *p* ≤ 0.05.

### 3.4. In Silico ADME Predictions

For the most promising compound **4f**, the free online platform SwissADME (http://www.swissadme.ch, accessed on 1 September 2022) [49] was used to predict absorption, distribution, metabolism, and elimination (ADME) properties, and also molecular descriptors related to Lipinski’s “Rule of Five” [22] and Veber’s extension [23], such as molecular weight (MW), octanol/water partition coefficient (*c*Log P), log of solubility in water (mol L^−1^) (Log *S*), number of hydrogen bond acceptors (HBA) and hydrogen bond donors (HBD), number of rotating connections (RB) and topological polar surface area (tPSA). The ability to inhibit or be a possible substrate for one of the five major isoforms of cytochrome P450 enzymes: CYP1A2, CYP2C9, CYP2D19, CYP2D6, and CYP3A4d, was predicted using the ADMETlab 2.0 (https://admetmesh.scbdd.com/service/evaluation/cal, accessed on 1 September 2022) [50]. Furthermore, site of metabolism predictions (SOMP) for the five major isoforms of cytochromes P450s were predicted using the site of metabolism prediction platform (http://www.way2drug.com/SOMP, accessed on 1 September 2022) [51].

### 3.5. Molecular Docking

#### 3.5.1. Proteins and Ligands Preparation for Docking

The structural models of ARG and TR from *L. amazonensis* were built by homology modeling with the aid of the SwissModel server (https://swissmodel.expasy.org/, accessed on 1 September 2022) [52]. The protein sequence of LaARG and LaTR was obtained from the UniProt database (accession number: O96394 and Q0GU43, respectively) [53]. The 3D structures used as a template were ARG of *L. mexicana* [32] (PDB: 4ITY, resolution: 1.80 Å) and TR from *Crithidia fasciculata* [33] (PDB ID: 1TYP, Resolution: 2.80 Å), obtained from the Protein Data Bank (PDB). These models were validated according to Camargo and co-workers [54].

The 3D structure of the most promising chalcone (**4f**) was built up in ChemDraw version 20.0.0.41 (PerkinElmer Informatics) [55]. The geometry optimization was then performed using the PM7 force field implemented in the MOPAC 2016 (Molecular Orbital Package) [56].

#### 3.5.2. Consensus Molecular Docking

Molecular docking was performed using AutoDock v. 4.2 [57] and AutoDock Vina [58]. The protein and ligand preparations were performed using AutoDockTools (ADT) v.1.5.6 [59]. All water molecules were removed, polar hydrogens atoms were added, and the Kollman and Gasteiger methods, respectively, assigned atoms with charges to protein and ligand. The center of the grid box from ARG for AutoDock was centered between Manganese ions at x: 15.141, y: −15.1248, z: −5.40, size 40 × 50 × 38 Å^3^ points, and spacing of 0.375 Å, for the AutoDock Vina the same coordinates were set, with a size of 20 × 23 × 19 Å^3^. For TR, the grid box for AutoDock was centered on the active binding site for T(S)_2_ at x: 71.856, y: 10.434, z: 12.967, size 60 × 60 × 60 Å^3^ points, and spacing of 0.375 Å, for AutoDock Vina, the size was set to 24 × 24 × 24 Å^3^. 

Molecular docking calculations were carried out considering the hybrid scoring function, implemented in Vina, and the genetic (GA) and Lamarckian genetic algorithms (LGA) [57], implemented in Autodock. The ligands performed 10 iterative runs and the best-scored pose for which docking resulted was considered the root-mean-square-deviation (RMSD) calculation.

Intermolecular interactions analysis and the RMSD calculations were carried out using Edu pyMOL version 1.7.4.4 [60]. The 2D graphics were generated using LigPlot+ v.2.2. version [61].

## 4. Conclusions

Nine 4,8-dimethoxynaphthalenyl chalcone derivatives **(4a**–**i)** were obtained from 15–85%. Among them, compounds **4b**, **4d**, **4e**, **4f**, **4g**, and **4i** had not been previously reported. Compound **4f** exhibited the best in vitro activity against promastigote forms of *L. amazonensis*. Regarding its mechanism of action, in vitro assays indicated that **4f** produced several morphological and ultrastructural cellular changes, increased intracellular ROS levels, and induced mitochondrial depolarization with selective cytotoxicity at non-toxic concentrations to mammalian cells. In addition, **4f** caused a reduction in infected macrophages and the number of amastigotes per cell, culminating in eliminating intracellular parasites. In silico studies of physicochemical and pharmacokinetic properties indicated that **4f** could present good oral bioavailability and intestinal absorption. The molecular docking study indicated that **4f** showed a greater affinity for ARG than TR. These findings are key to revealing promising molecules for new antileishmanial therapies based on ARG inhibition, directing us to further drug design studies aimed at optimization by structural modifications to provide a more active analog with better pharmacokinetic properties.

## Data Availability

The cif-file for compound **4i** has been deposited with the Cambridge Crystallographic Data Centre with deposition number, 1913905, copies of which can be obtained free of charge on written application to CCDC, 12 Union Road, Cambridge, CB2 1EZ, UK (fax: +44 1223 336033); on request by e-mail to deposit@ccdc.cam.ac.uk or by access to http://www.ccdc.cam.ac.uk (accessed on 1 September 2022). Additional data used to support the findings of this study are available from the corresponding author upon request.

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
