# Peer review of "Antileishmanial Activity of 4,8-Dimethoxynaphthalenyl Chalcones on Leishmania amazonensis"

_antibiotics, 2022, doi:10.3390/antibiotics11101402_

Round 1
Reviewer 1 Report
The manuscript presents the rational design, synthesis, and biological evaluation of against promastigote and amastigote forms of Leishmania amazonensis of nine 4,8-dimethoxynaphthalenyl chalcones. This study could be very useful for the new preparation Antileishmanial compounds. The study is interesting and suitable for the publication, but there are some comments regarding the manuscript. Authors may find useful to consider following comments and suggestions in preparation of the manuscript.
1. Some minor errors related to the language must be corrected.
2. The important structures with Antileishmanial activity results must be added to the “introduction” section for reflecting the importance of the work and history.
3. The number of molecules presented in the scheme 1 is not enough to come out with SAR. The position and type of substituent is highly important; i.e. can anyone tell me that trimethoxy substituted chalcones cannot inhibit or vice versa. That must be enlarged.
4. The structure elucidation has been performed by using different NMR techniques (COSY, HSQC NMR etc). The authors must show the readers the reason of usage of these techniques with figures. The authors must also explain the characteristic peaks in proton and carbon NMR as well as in FTIR.
5. I am curious about the BBB permeability results since the most active molecule has no BBB permeability.
6. 5 of 9 molecules are known and reported before. So, the melting points in the literature must be reported of known molecules.
7. Thoroughly check the consistency of references.
Author Response
Dear reviewer, the manuscript was revised and the modifications are indicated below and highlighted with a yellow background in the manuscript.
Some minor errors related to the language must be corrected.
Authors: Dear reviewer, we have thoroughly checked the wording of our manuscript.
The important structures with Antileishmanial activity results must be added to the “introduction” section for reflecting the importance of the work and history.
Authors: Dear reviewer, we have included structures with the chalcone nucleus recently reported in the literature with antileishmanial activity, as suggested.
The number of molecules presented in the scheme 1 is not enough to come out with SAR. The position and type of substituent is highly important; i.e. can anyone tell me that trimethoxy substituted chalcones cannot inhibit or vice versa. That must be enlarged.
Authors: Dear reviewer, we believe that the synthesis of new molecules is out of the scope of the present work. We have designed and synthesized several chalcone analogues, and a robust SAR study has been presented, where different substituents with diverse properties have been studied. Most part of the derivatives are new and extensive characterization data have been provided, with crystal structure, NMR, mass spectrometry and infrared. Several biological assays have been performed and a new hit with excellent activity and low toxicity was discovered. In this way, we believe that innovative and relevant results have been described in the present work and the synthesis of new molecules can be more productive in a subsequent step of our research, where more specific modifications can be proposed from our new hit.
The structure elucidation has been performed by using different NMR techniques (COSY, HSQC NMR etc). The authors must show the readers the reason of usage of these techniques with figures. The authors must also explain the characteristic peaks in proton and carbon NMR as well as in FTIR.
Authors: Dear reviewer, the cited NMR techniques were necessary for unequivocal correlation of the signals observed in the spectra with the respective protons and carbons. We believe that a discussion of how the correlation was done would leave the already large article extensively large and not add really informative content to the reader. All correlations are presented in the experimental section, in addition, additional techniques such as XRD helped us for an unequivocal structure determination.
I am curious about the BBB permeability results since the most active molecule has no BBB permeability.
Authors: Dear reviewer, we understand that promising compounds intended for the treatment of leishmaniasis that can permeate the blood-brain barrier are not desirable, as they may be responsible for CNS adverse effects, given that the clinical manifestations of leishmaniasis do not directly affect the central nervous system such as African trypanosomiasis, for example.
5 of 9 molecules are known and reported before. So, the melting points in the literature must be reported of known molecules.
Authors: Dear reviewer, only 3 are known. We have corrected this information in our manuscript. Melting points have already been reported in the "Materials and Methods" section, along with the spectroscopic information. The melting points available in the literature were duly mentioned. The others were determined according to the methodology available in the "Materials and Methods" section: 4a (line 479), 4b (line 488), 4c (line 498), 4d (line 508), 4e (line 518), 4f (line 529), 4g (line 539), 4h (line 550), 4i (line 559).
Thoroughly check the consistency of references.
Authors: Dear reviewer, we have thoroughly checked all references as suggested.
Reviewer 2 Report
The present manuscript entitled "Antileishmanial Activity of 4,8-dimethoxynaphthalenyl Chalcones on Leishmania amazonensis" by Kaio Maciel Santiago-Silva, Bruna Taciane da Silva Bortoleti, Laudicéa do Nascimento Oliveira, Fernanda Lima de Azevedo Maia, Joyce Cristina Castro, Ivete Conchon Costa, Danielle Bidóia Lazarin, James Lewis Wardell, Solange M.S.V. Wardell, Magaly Girão Albuquerque, Camilo Henrique da Silva Lima, Wander Rogério Pavanelli, Marcelle de Lima Ferreira Bispo, and Raoni Schroeder Borges Gonçalves (antibiotics-1921536) is written correctly and has a good structure; moreover, it has all the necessary parts. The article is interesting from a medical point of view; therefore, it should interest the reader. I proposed improvements in the method description and with a presentation of figures. The paper meets Antibiotics' requirements, and I recommend the article for publication in Antibiotics following the common editing stage. My current decision is a minor revision. More specific comments and observations are presented below.
1. Please check whether the use of bolds in the text complies with the journal's requirements.
2. Figure 1. This figure is not mentioned in the text. The figure should be described in the article. The quality of this drawing, especially its text, also needs to be improved.
3. Page 3, line 91. References should appear in ascending order. Please renumber and check other items.
4. The Celsius degree sign looks strange in some places and needs to be corrected.
5. Page 4, line 107. "Fig. 2b" is mentioned, but in the drawing, no distinction is made between "a" and "b".
6. The drawings are sometimes distinguished by lower case letters "a", "b" and sometimes capital letters "A", "B". This should be unified.
7. Page 3, line 119. It should be "Fig. 3" instead of "Fig. 4".
8. Figure 3. “4f” is invisible.
9. Page 6, lines 147 and 148. The “relationship” is mentioned. This term should be changed to "relation". The relationship tends to be used more broadly to describe the interactions between specific people or smaller groups of people.
10. Figure 7. The y-axis description is too large.
11. Table 2. Part of the text is not visible.
12. Materials and methods. The parameters used during the analysis should be mentioned in more detail for all measurement techniques. There is no mention of the reagents together with the company and country of origin used for the syntheses. Please add more information about flash chromatography. Appropriately substituted benzaldehyde should be listed.
13. Appropriate tools should be used to characterize the synthesis method (e.g., AGREE - Analytical GREEnness Metric Approach).
14. The text includes websites that should be mentioned as references, while the References section should also contain access dates.
15. Page 20, line 623. The notation of the unit differs from the previously presented provisions. Please check and standardize the record of units.
16. Does the developed studies have disadvantages?
17. Conclusions. Please, emphasize clearly the advantages of the research carried out.
18. References. [5] and [6] are the same.
I hope that the comments presented will help improve the article.
Author Response
Dear reviewer,
The manuscript was revised and the modifications are indicated below and highlighted with a yellow background in the manuscript.
Please check whether the use of bolds in the text complies with the journal's requirements.
Authors: Dear reviewer, in a new consultation in the “Instructions to Authors”, we did not find any requirement regarding the use of bold.
Figure 1. This figure is not mentioned in the text. The figure should be described in the article. The quality of this drawing, especially its text, also needs to be improved.
Authors: Dear reviewer, we mention the “Figure 1” in the text as requested. We adapted the quality according to the magazine’s specifications (minimum 1000 pixels in width/height, or resolution of 300 dpi or higher), and all figures were sent in a single zip file according to the “Instructions to Authors”.
Page 3, line 91. References should appear in ascending order. Please renumber and check other items.
Authors: Dear reviewer, we have corrected the order of references in our manuscript.
The Celsius degree sign looks strange in some places and needs to be corrected.
Authors: Dear reviewer, we have corrected the Celsius degree symbol (℃) as requested.
Page 4, line 107. "Fig. 2b" is mentioned, but in the drawing, no distinction is made between "a" and "b".
Authors: Dear reviewer, we distinguish Figure 2 between “A” and “B” as requested.
The drawings are sometimes distinguished by lower case letters "a", "b" and sometimes capital letters "A", "B". This should be unified.
Authors: Dear reviewer, we only use capital letters “A”, “B” to distinguish figures as requested.
Page 3, line 119. It should be "Fig. 3" instead of "Fig. 4".
Authors: Dear reviewer, we corrected the mention of figure 3 in the text.
Figure 3. “4f” is invisible.
Authors: Dear reviewer, we corrected compound 4f in Figure 3 in our manuscript.
Page 6, lines 147 and 148. The “relationship” is mentioned. This term should be changed to "relation". The relationship tends to be used more broadly to describe the interactions between specific people or smaller groups of people.
Authors: Dear reviewer, we appreciate the suggestion, but the word “relationship” present in lines 162-163 is related to the context of the “Structure-Activity Relationship (SAR) strategy largely applied in drug discovery. According to the International Union of Pure and Applied Chemistry (IUPAC), SAR is defined as the “Association between specific aspects of molecular structure and defined biological action”. Therefore, the word “relationship” was properly used in this sentence.
Reference: UPAC. Compendium of Chemical Terminology, 2nd ed. (the "Gold Book"). Compiled by A. D. McNaught and A. Wilkinson. Blackwell Scientific Publications, Oxford (1997). Online version (2019-) created by S. J. Chalk. ISBN 0-9678550-9-8. https://doi.org/10.1351/goldbook.
Figure 7. The y-axis description is too large.
Authors: Dear reviewer, we have shortened the description of the y-axis in Figure 7 as requested.
Table 2. Part of the text is not visible.
Authors: Dear reviewer, we framed Table 2 to make the entire text visible.
Materials and methods. The parameters used during the analysis should be mentioned in more detail for all measurement techniques. There is no mention of the reagents together with the company and country of origin used for the syntheses. Please add more information about flash chromatography. Appropriately substituted benzaldehyde should be listed.
Authors: Dear reviewer, the information was included in the manuscript.
Appropriate tools should be used to characterize the synthesis method (e.g., AGREE - Analytical GREEnness Metric Approach).
Authors: Dear reviewer, we have checked, and the Analytical GREEnness is used for the evaluation of analytical methods, which would not fit in the present work. If the reviewer has any suggestions of free programs for the evaluation of the synthesis, we would be grateful.
The text includes websites that should be mentioned as references, while the References section should also contain access dates.
Authors: Dear reviewer, we mentioned the sites as references (45 − 48) and added the access date.
Page 20, line 623. The notation of the unit differs from the previously presented provisions. Please check and standardize the record of units.
Authors: Dear reviewer, we have verified and standardized the unit records in our manuscript as requested.
Does the developed studies have disadvantages?
Authors: Dear reviewer, we do not understand this as a disadvantage, but a limitation so far is that our findings have not been consolidated through in vivo assays. In further studies, we intend to verify the potential of 4f in L. amazonensis infection models.
Conclusions. Please, emphasize clearly the advantages of the research carried out.
Authors: Dear reviewer, we emphasize in our conclusion the advantages of the research carried out.
References. [5] and [6] are the same.
Authors: Dear reviewer, we have corrected these references in our manuscript.
Round 2
Reviewer 1 Report
The compound is not new, therefore it must be cited with the reference.
The author have revised manuscript carefully except that citation of compound 4a. After addition of reference, then, it can be accepted.
Author Response
reviewer #1 The compound is not new, therefore it must be cited with the reference.The author have revised manuscript carefully except that citation of compound 4a. After addition of reference, then, it can be accepted.
Authors: Dear reviewer, as requested, compound 4a was duly cited in the "Materials and Methods" section, just after the melting point (line 432).